# Electrohydrodynamic Printed Ultra-Micro AgNPs Thin Film Temperature Sensors Array for High-Resolution Sensing

**DOI:** 10.3390/mi14081621

**Published:** 2023-08-17

**Authors:** Yingping He, Lanlan Li, Zhixuan Su, Lida Xu, Maocheng Guo, Bowen Duan, Wenxuan Wang, Bo Cheng, Daoheng Sun, Zhenyin Hai

**Affiliations:** Department of Mechanical and Electrical Engineering, Xiamen University, Xiamen 361005, China; 18750605740@163.com (Y.H.); lilanlan1006@163.com (L.L.); 19920221151557@stu.xmu.edu.cn (Z.S.); superldxu@gmail.com (L.X.); gmcxmu@163.com (M.G.); 19920211151508@stu.xmu.edu.cn (B.D.); 19920221151600@stu.xmu.edu.cn (W.W.); chengperfect@163.com (B.C.)

**Keywords:** electrohydrodynamic printing, ultra-micro, thin film, temperature sensors array, AgNPs

## Abstract

Current methods for thin film sensors preparation include screen printing, inkjet printing, and MEMS (microelectromechanical systems) techniques. However, their limitations in achieving sub-10 μm line widths hinder high-density sensors array fabrication. Electrohydrodynamic (EHD) printing is a promising alternative due to its ability to print multiple materials and multilayer structures with patterned films less than 10 μm width. In this paper, we innovatively proposed a method using only EHD printing to prepare ultra-micro thin film temperature sensors array. The sensitive layer of the four sensors was compactly integrated within an area measuring 450 μm × 450 μm, featuring a line width of less than 10 μm, and a film thickness ranging from 150 nm to 230 nm. The conductive network of silver nanoparticles exhibited a porosity of 0.86%. After a 17 h temperature-resistance test, significant differences in the performance of the four sensors were observed. Sensor 3 showcased relatively superior performance, boasting a fitted linearity of 0.99994 and a TCR of 937.8 ppm/°C within the temperature range of 20 °C to 120 °C. Moreover, after the 17 h test, a resistance change rate of 0.17% was recorded at 20 °C.

## 1. Introduction

Temperature sensors, widely employed across diverse fields including industrial and agricultural production, biomedical applications, aerospace engineering, and numerous other domains, play a vital role in monitoring and controlling temperature variations [1,2,3,4]. In recent years, the temperature sensors array has garnered significant attention, particularly in high-temperature applications, due to their capability to provide real-time temperature distribution at specific locations and subsequently infer the temperature field distribution across the entire space. Increasing the density of temperature sensors is crucial for improving the accuracy and precision of computational temperature field distribution. This particular requirement necessitates the utilization of temperature sensors characterized by ultra micro dimensions, enabling easy deployment and placement within the desired area. Traditional temperature sensors, such as thermocouples for blocks, often occupy a significant amount of space, making it challenging to arrange them in a compact manner [5,6,7]. Therefore, these sensors typically provide average temperature measurements for large spaces and cannot achieve real-time monitoring of temperature fields in limited areas. Compared to traditional temperature sensors, thin film temperature sensors offer several advantages, including fast response, compact size, lightweight design, high sensitivity, and in situ preparation [8,9,10,11]. Therefore, an array of ultra-micro thin film temperature sensors can be utilized to measure the temperature of multiple small surfaces within a confined space. This enables high-resolution and real-time monitoring of the temperature field.

Currently, the main techniques utilized for fabricating thin film sensors arrays include screen printing, inkjet printing, and MEMS technique. Indeed, screen printing is a widely utilized method for pattern preparation and has found extensive application in the field of flexible sensing [12,13,14]. It offers several advantages, including a straightforward process, high efficiency, cost-effectiveness, and suitability for mass production. In this technique, a paste is deposited onto a stencil, and the desired pattern is transferred onto the substrate through the stencil by applying pressure. The accuracy of pattern printing in screen printing is closely influenced by factors such as the number of printing plate grids, the type of material used, and the solvent of the paste [15]. Currently, the maximum achievable printing accuracy is approximately 100 μm [16]. Inkjet printing technology is a type of drop-on-demand inkjet technology, most commonly driven by thermal or piezoelectric methods, known as piezoelectric inkjet and thermal bubble inkjet, respectively. It has been widely adapted for applications in electronics, optics, bioengineering, and other areas [17,18,19]. The pulse pressure acts on the ink, which is passed through tiny nozzles and then deposited non-contactingly on a substrate with high control. A programmable mobile platform can be coordinated with it to create patterned printing. One key challenge in inkjet printing is achieving practical levels of resolution. On the one hand, the ink needs to be deposited onto the substrate through small nozzles, making the resolution highly dependent on the nozzle diameter. However, the smallest commercially available inkjet nozzles currently have a diameter of around 20 μm. On the other hand, the printed droplets tend to spread and diffuse within the substrate, with even more significant diffusion occurring during multi-layer printing. In fact, the fabrication of sensors often requires multi-layer printing to achieve specific functionalities. Due to the limitations of inkjet printing technology, it is currently impossible to achieve a sensitive layer with a linewidth of 10 μm. The diffusion and spreading of printed droplets, along with the minimum nozzle diameter of currently available inkjet printers, make it challenging to achieve such fine resolution. However, the line width of thin film sensors produced by techniques such as screen printing or inkjet printing is incapable of reaching 10 μm, rendering it unsuitable for creating ultra-micro sensors arrays. Most ultra-micro sensors arrays are fabricated using MEMS technology. These sensors find applications in various fields such as health monitoring, gas detection, pressure measurement, and biochips. For example, Hardik J Pandya et al. developed a flexible MEMS-based electro-mechanical sensor array for breast cancer diagnosis [20]. The sensor array covers an area of 180 μm × 180 μm. A G P Kottapalli et al. created a flexible MEMS pressure sensor array using liquid crystal polymer (LCP) for fish-like underwater sensing [21]. The array consists of ten sensors with dimensions of 60 mm (L) × 25 mm (W) × 0.4 mm (H). Roy M. Pemberton et al. designed a micro(bio)sensor array chip for simultaneous measurements of important cell biomarkers [22]. Michael Blaschke et al. developed a gas-sensor array based on MEMS technology for monitoring the perceived air quality inside car cabins [23]. However, the process of patterned deposition of micro-nano films using MEMS technology involves the use of complex and precise masks. Consequently, this results in high production costs, extended processing times, and limitations in the choice of target materials and the preparation of curved surfaces [24,25]. 

In comparison, additive manufacturing (AM) technology has the advantage of directly depositing multi-layered structures and ultra-micro thin films. It also offers significant cost-effectiveness, the potential for large-scale production, and a reduced environmental impact [26]. In the past decade, considerable efforts have been dedicated to reducing the printing resolution from micrometers to nanometers, which is crucial for advancements in printed electronics technology [27]. Among the various additive manufacturing (AM) technologies, EHD printing stands out as a technique capable of depositing thin films with narrow line widths and achieving micro-nano resolution [28]. This technology operates by applying electric field forces that induce tangential stress on the ink surface, causing the hemispherical surface at the nozzle tip to deform into a cone shape known as a Taylor cone, with a semi-vertical angle of 49.3° [29]. The Taylor cone exists at the boundary between stability and instability. By slightly increasing the voltage, the electrostatic stress surpasses the combined surface tension and viscosity of the ink surface, resulting in the generation of a jet. This phenomenon enables the production of droplets or ink jets that are two to five orders of magnitude smaller than the nozzle size [30]. In summary, the investigation of EHD printing for fabricating multi-layered structures and ultra-thin film temperature sensors hold significant implications for achieving high-density and arrayed temperature sensing within constrained spaces. It reduces the cost and time required for preparing micro-thin film sensors and expands the range of functional materials that can be utilized. 

Due to the current limitations in the selection of printable ink materials, micro-temperature sensors prepared using EHD printing technology, such as AgNPs (Silver Nanoparticles), exhibit significant insufficient in terms of certain performance aspects, such as temperature resistance, when compared to temperature sensors fabricated from other materials like SiC [31], Ni/4H-nSiC [32,33], TiO_2_ [34], and BiFeO_3_/TiO_2_ [35]. However, the temperature sensors manufactured using EHD printing technology show promise in achieving miniaturization in the production of micro-temperature sensors. Currently, there are limited reports on the fabrication of micro thin-film temperature sensor arrays using EHD technology. Even for individual thin-film temperature sensors, the sensitive area typically surpasses 10 mm^2^, with a minor portion falling within the range of 1 mm^2^ to 2 mm^2^ [26,36,37]. The line width of these sensors predominantly ranges between 50 μm and 100 μm [26,36,37]. Waqas Kamal et al. employed EHD technology to print nanosilver film temperature sensors on a flexible PET substrate [26]. The sensor’s sensitive area was around 3500 μm × 5000 μm, with a line width of 90 μm. Their results demonstrated a TCR of 3400 ppm/°C and an upper temperature measurement limit of 110 °C. Kyung Hyun Choi et al. utilized a unique Roll-to-Roll (R2R) system to EHD print nanosilver film temperature sensors on a flexible PET substrate, with a sensitive area of approximately 1100 μm × 1200 μm and a minimum line width of 30 μm [36]. Their study focused on the single-step heating characteristics of resistance, reporting a TCR of 768.7 ppm/°C and an upper limit of 105 °C for temperature measurement. Salman Ahmad et al. applied EHD printing technology to fabricate nanosilver film temperature sensors on a rigid glass substrate [37]. The sensors possessed a sensitive area of about 5000 μm × 5000 μm and a line width of 50 μm. Their research revealed that the sensors exhibited a TCR of 11,500 ppm/°C, an upper temperature measurement limit of 100 °C, and displayed significant hysteresis and drift when subjected to multi-cycle resistance measurements. 

This article focuses on ultra-micro thin film temperature sensors array fabricated by EHD printing technique, choosing AgNPs-based ink as the sensitive material, to study the preparation and performance of the sensors array. Compared to previous research, this temperature sensors array has smaller line widths, measuring less than 10 μm, and achieves a higher level of integration. By employing the EHD printing method, the sensitive layer of four sensors was precisely deposited within a confined area not exceeding 450 μm × 450 μm. The average line width achieved was maintained below 10 μm, while the film thickness ranged between 150 nm and 230 nm. Subsequently, lead wires were printed to ensure electrical conductivity and optimize the printing efficiency, employing comparatively wider line widths. Lastly, a diluted PDMS (polydimethylsiloxane) material was printed to effectively encapsulate the sensor array, providing a PDMS encapsulation. After a 17 h temperature resistance test, the performance test results showed that the best sensors exhibited low hysteresis with high repeatability, despite poor sensor performance uniformity in the sensor array. The EHD method eliminated the need for expensive equipment and precise masks in the fabrication of high-density sensor arrays, overcoming the large line width or expensive preparation costs of typical preparation methods. This method holds tremendous potential for achieving high-density, arrayed thin film sensors, particularly on curved surfaces. 

## 2. Materials and Methods

### 2.1. Materials

In this study, a double-sided polished sapphire wafer with a thickness of 600 μm was employed as the substrate for the sensing unit, which was provided by Shanghai Xuanyisheng Technology Co., Ltd. (Shanghai, China). The ink used for printing contained approximately 30–35 wt% solid content of AgNPs, with a viscosity of around 20 cp. Prior to printing, the ink was subjected to 60 min of ultrasonic treatment to ensure the uniform dispersion of suspended silver nanoparticles in the solution. 

The encapsulation layer primarily consisted of polydimethylsiloxane (PDMS). A two-part PDMS solution, obtained from Dow Consumer (Midland, MI, USA) (product number SYLGARD 184), was utilized, comprising a basic component and a silicon-oil-based curing agent, mixed in a mass ratio of 10:1. N-hexane, obtained from Macklin (Shanghai, China), served as a PDMS diluent. A mixture of n-hexane and PDMS in a 3:1 ratio was used as the EHD printing ink. 

### 2.2. EHD Printing System

The schematic diagram of our self-built EHD printing system is depicted in Figure 1. The main components of the printing system included a programmable 3D motion platform, voltage amplifier, signal generator, oscilloscope, camera, and light source. Other supplementary devices, such as a heating plate with precise temperature control and a nozzle fixture, are not shown in the figure. 

The accurate positioning of the substrate was accomplished using a three-dimensional motion platform with a resolution of 1.25 μm in the X and Y axes and 0.375 μm in the Z axis. The function generator (Tektronix TBS-1102) and oscilloscope (RIGOL DG1022Z) were connected to the voltage amplifier, which can provide a maximum voltage of 4 kV. The Tektronix TBS-1102 is manufactured by Tektronix, Inc., headquartered in Beaverton, Oregon, United States. The RIGOL DG1022Z is manufactured by RIGOL Technologies, and the company is headquartered in Beijing, China. The function generator generated the required voltage signal waveform, while the oscilloscope monitored the amplified voltage waveform. The glass tip used in our system was custom-made using a glass drawing machine. Unlike a metal tip, the glass tip offered significant advantages, including prevention of near-field breakdown and enabling microfluid supply. The nominal end tip diameter of the glass tip was 2 μm, with a tolerance of ±0.5 μm, in contrast to the current metal tip with a diameter of approximately 50 μm. To observe the droplet and printing behavior at the nozzle tip, a camera and LED light source were appropriately positioned. 

### 2.3. Fabrication Process

The procedure for fabricating ultra-micro AgNPs temperature sensor arrays with patterned, high-density, and in situ characteristic using only EHD printing technology, as shown in Figure 2. Figure 2a illustrates the surface treatment process of the substrate, including ethanol ultrasonic cleaning, air gun dust removal, and plasma surface treatment. The sapphire substrate was ultrasonically cleaned with ethanol and dried in an 80 °C oven for 5 min. After removal, the surface was cleaned with an air gun to ensure that there was no dust on the surface. Finally, a plasma surface treatment was performed to increase surface hydrophilicity and reduce ink contact angle. 

Figure 2b demonstrates the printing process of the sensitive layer for the four sensors, with each sensor’s sensitive layer printed 20 times. Figure 2c showcases the printing process of the AgNPs conductive lead, wherein the lead layer of each sensor was printed six times using a higher voltage. Figure 2d illustrates the curing process of the sensor array at 120 °C for a duration of 10 min, preparing it for the subsequent PDMS encapsulation by EHD printing. Figure 2e demonstrates the EHD printing process of six layers of PDMS for encapsulating the sensor arrays, effectively preventing oxidation of silver during temperature resistance testing. Figure 2f visually depicts the preparation of solder joints and the subsequent sensor sintering process, meticulously conducted at a controlled temperature of 140 °C for a duration of 1 h. The optical microscope images of the sensor arrays without encapsulation are shown in Figure 3a–c, which were fabricated by EHD printing. The printed trajectory was continuous and had a uniform line-width. 

### 2.4. Measurements

Figure 4a shows the test platform used to test the resistance-temperature data, and the accurate temperature was obtained by an adhesive thin film T-type thermocouple. A constant temperature and humidity chamber was utilized, enabling the control of both temperature and humidity over time. Throughout the experiments, a constant humidity level of 0% was maintained, while the temperature varied with time. The four-wire configuration test circuit diagram is shown in Figure 4c. The resistance of the RTD was calculated by dividing the measured voltage by the supplied excitation source current [38]. 

## 3. Results and Discussion

### 3.1. Morphological and Physical Characterizations

For the sensors array without encapsulation, scanning electron microscopy (SEM) was used to characterize the size of the ultra-micro AgNPs thin film sensitive grids and the agglomeration of nanoparticles. As shown in Figure 5, SEM pictures of the sensors array printed on a sapphire substrate are given. Figure 5a,c shows that four sensitive grids of the sensor array were the four-wire configuration, and the four sensor grids were integrated within a range not exceeding 450 μm × 450 μm, with a linewidth of less than 10 μm. As shown in Figure 5b,d,f, the pronounced reactivity of the Ag element and its distribution range consistent with the SEM testing results provided evidence that the conductive network was composed of Ag particles. 

Figure 6a shows the SEM image of the dashed region in Figure 5e. It can be observed that the majority of the Ag nanoparticles exhibited a uniform size distribution, with only a small amount sintering agglomeration and the presence of pores. The PCAS software (version 2.324)was utilized to validate the reported porosity and compactness characteristics of silver nanoparticles. PCAS is a specialized software used for the identification and quantitative analysis of pore systems and fracture systems. This advanced tool can automatically recognize various pores and fractures in the image, providing various geometric and statistical parameters. From Figure 6b, the original image was transformed into a binary image, where the black regions represent non-porous structures, and the white regions indicate possible presence of pores. The result image obtained from the specific algorithmic calculation is shown in Figure 6c, where the black regions were identified as non-porous structures, and the remaining parts were considered to contain pores. After using the PCAS software to calculate the porosity, it was determined to be 0.86%. The dense and pore-free nature of the silver particle-based film layer confirmed the conductivity of the ultra-micro sensors array. 

To demonstrate the line width and film thickness of the sensor array, the midpoints (Figure 5c) of the four sensitive grinds in the sensor array were tested at a total of 16 positions using AFM. For the film thickness data, as depicted in Figure 7a, the thickness at the four positions of Sensor 1 was approximately 203 nm, 217 nm, 230 nm, and 222 nm, with an average of 218 nm. Similarly, the thickness at the four positions of Sensor 2 was approximately 213 nm, 226 nm, 217 nm, and 219 nm, with an average of 218.75 nm. The thickness at the four positions of Sensor 3 was approximately 166 nm, 196 nm, 179 nm, and 198 nm, with an average of 184.75 nm. Lastly, the thickness at the four positions of Sensor 4 was approximately 157 nm, 159 nm, 157 nm, and 150 nm, with an average of 155.75 nm. For the data obtained from AFM testing, there is currently no standard method to define line width, especially for micro line width with particle diffusion at the edges. By combining SEM images with AFM three-dimensional images and excluding the effects of particle diffusion, the boundary for line width calculation was determined to be the boundary of thin films with a thickness of 50 nm or more. This was because 50 nm happened to be the height of a single Ag particle. As depicted in Figure 7a, the linewidth at the four positions of Sensor 1 was approximately 8.373 μm, 8.510 μm, 9.196 μm, and 7.686 μm, with an average of 8.441 μm. As shown in Figure 7b, the linewidth at the four positions of Sensor 2 was approximately 8.647 μm, 9.471 μm, 8.647 μm, and 8.235 μm, with an average of 8.75 μm. In Figure 7c, the linewidth at the four positions of Sensor 3 was approximately 6.588 μm, 8.373 μm, 7.824 μm, and 7.412 μm, with an average of 7.55 μm. Finally, in Figure 7d, the linewidth at the four positions of Sensor 4 was approximately 7.275 μm, 6.726 μm, 7.245 μm, and 6.588 μm, with an average of 6.96 μm. The results indicate that the linewidth of the sensitive layer in this sensor array was within 10 μm, and the film thickness distribution ranged between 150 nm and 230 nm. This also confirmed that the sensor array was sufficiently thick to establish a conductive network. The dimensionless values obtained by dividing the average linewidth of the sensor by the average film thickness were 38.72, 40.00, 40.87, and 44.69, respectively, revealing variations in film thickness and linewidth consistency among the sensors. The observed variations in film thickness and linewidth among the sensors revealed the significant impact of the stopping voltage when printing the four sensitive layers. However, when considering individual sensor film thickness and linewidth parameters, there was strong control over linewidth and film thickness capabilities. 

### 3.2. Electrical Characterization 

The sensor array underwent a 17 h test in chamber, during which a round of stepped temperature resistance tests was conducted. The resulting data from the stepped temperature resistance tests are depicted in Figure 8a,c,e,g, showcasing the variation in resistance rates of individual sensors and the temperature changes measured by thermocouples over time. Data points from the stepped temperature tests were used to plot resistance change rate curves, as depicted in Figure 8b,d,f,h, within the temperature range of 20 °C to 120 °C. This procedure facilitated the subsequent assessment of the resistance change rate over a 17 h period, linearity during the heating stage, and TCR for each sensor within the sensor array. In Figure 8b, it is shown that for sensor 1, the TCR within the temperature range of 20 °C to 120 °C was 883.4 ppm/°C, with a fitted linearity of 0.9996. Additionally, after the 17 h test, the resistance change rate at 20 °C was 0.66%. In Figure 8d, it is shown that for sensor 2, the TCR within the temperature range of 20 °C to 120 °C was 720.0 ppm/°C, with a fitted linearity of 0.9995. Additionally, after the 17 h test, the resistance change rate at 20 °C was below 0.056%. In Figure 8f, it is shown that for sensor 3, the TCR within the temperature range of 20 °C to 120 °C was 937.8 ppm/°C, with a fitted linearity of 0.99994. Additionally, after the 17 h test, the resistance change rate at 20 °C was 0.17%. In Figure 8h, it is shown that for sensor 4, the TCR within the temperature range of 20 °C to 120 °C was 527.2 ppm/°C, with a fitted linearity of 0.9993. Additionally, after the 17 h test, the resistance change rate at 20 °C was 0.81%. Compared to the other three sensors, sensor 3 was regarded as having relatively superior performance. Although its resistance change rate was higher than that of sensor 2, its linearity and TCR surpassed those of the other sensors, and it exhibited minimal hysteresis. Additionally, for short-duration measurements, the 0.01%/h (0.17% divided by 17 h) resistance change rate had a negligible impact.

The significant rate of resistance change observed at 20 °C after the 17 h test provides evidence of the occurrence of oxidation (also known as corrosion) in silver nanoparticles. Sensors exhibiting a pronounced resistance change rate may potentially be influenced by non-uniform PDMS encapsulation. According to relevant literature, silver nanoparticle suspensions can be stored for extended periods, but they corrode rapidly when exposed to the atmosphere, as demonstrated in studies [39]. The first investigation found degradation within hours of exposure to laboratory air, detected through shifts in surface plasmon response and confirmed to involve sulfur in the corrosion product [40]. Another study affirmed similar degradation to a sulfur-containing product in air, while corrosion was absent in vacuum-stored nanoparticles [41]. The appearance of the corrosion product resembled particulates seen in bulk corrosion. Concurrently, an analysis of solution-synthesized silver particles exposed to air identified the final corrosion product as silver sulfide [42]. Due to the presence of silver sulfide, the electronic transmission within the Ag conductive network was impeded, resulting in an increase in thin film resistance. 

The differences in resistance change rates among various sensors can also be referred to as differences in TCR. In essence, despite the four sensors seemingly being the same, there were slight distinctions in the manufacturing process. These differences were very subtle and micro, and controlling them under such extreme conditions is challenging. The significant TCR differences observed among the four sensors might be due to the influence of voltage during the printing of different sensor layers. Specifically, while printing the sensitive layer of the first sensor, the high voltage was reduced to adjust droplet volume. This process ensured a concentration of Ag particle droplets for subsequent printing. However, when printing the sensitive layer of the second sensor, stopping the voltage was necessary, and it was not possible to fine-tune the droplet volume (excessive volume could damage the previously printed sensors due to their close proximity). Consequently, adjustments were usually made based on the voltage settings used for the first sensor. This printing method significantly increased the proportion of solvent, leading to a decrease in TCR. On a larger scale, increasing the line width can mitigate these micro-level process differences, thereby enhancing TCR consistency. The primary focus of this paper was to demonstrate the feasibility of an ultra-micro thin film temperature sensors array under extreme linewidth conditions. Subsequent work will decrease the line width limitation for flexible applications. 

## 4. Conclusions

In conclusion, the ultra-micro AgNPs thin film temperature sensors array with patterned, high-density, and in situ characteristics was successfully fabricated solely using EHD printing. SEM results confirmed the integration of the four sensor grids within an area not exceeding 450 μm × 450 μm, featuring line widths below 10 μm. EDS results validated the conductive network’s composition as Ag particles. Further assessment indicated a porosity of 0.86% in the conductive network, confirming its reliability of the conductive network. AFM results also revealed an average line width below 10 μm in the sensor array’s sensitive layer, with a minimum of 6.6 μm, and a membrane thickness ranging from 150 nm to 230 nm. After a 17 h temperature-resistance test, significant differences in the performance of the four sensors were observed. Among these sensors, sensor 3 demonstrated relatively better performance with a fitted linearity of 0.99994, featuring a TCR of 937.8 ppm/°C within the temperature range of 20 °C to 120 °C. Furthermore, following the 17 h test, a resistance change rate of 0.17% was recorded at 20 °C. This result suggests the possibility of non-uniform thickness in the EHD-printed PDMS encapsulation layer, which might have contributed to the inability to effectively prevent oxidation of the sensor array. This study marks a preliminary achievement in demonstrating the feasibility of utilizing EHD printing for fabricating an ultra-micro AgNPs thin film temperature sensor array. In the future, the application and encapsulation of such sensors on curved surfaces will be explored. 

## Figures and Tables

**Figure 1 micromachines-14-01621-f001:**
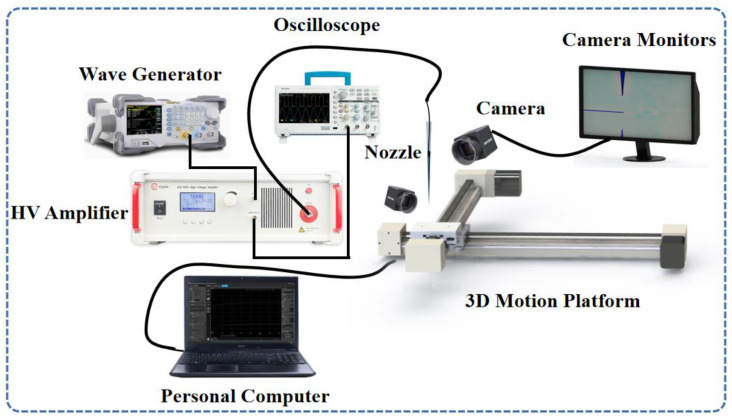
Schematic illustration of a drop-on-demand EHD jet printer.

**Figure 2 micromachines-14-01621-f002:**
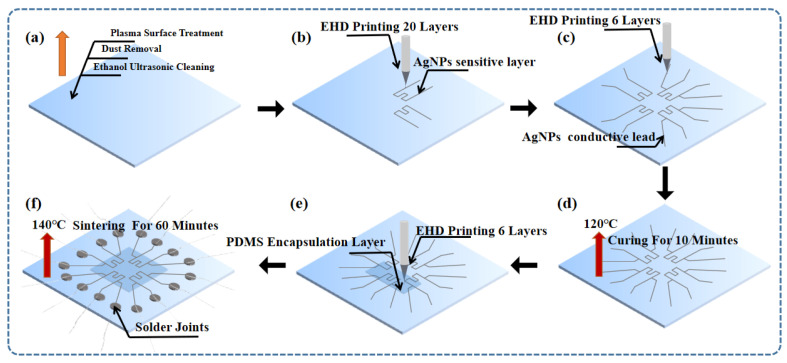
(**a**) Substrate treatment processes. (**b**) EHD printing sensitive layer process. (**c**) EHD printing electrical lead process. (**d**) Sensor arrays curing process. (**e**) EHD printing encapsulation layer process. (**f**) Sensor sintering and solder joint curing process.

**Figure 3 micromachines-14-01621-f003:**
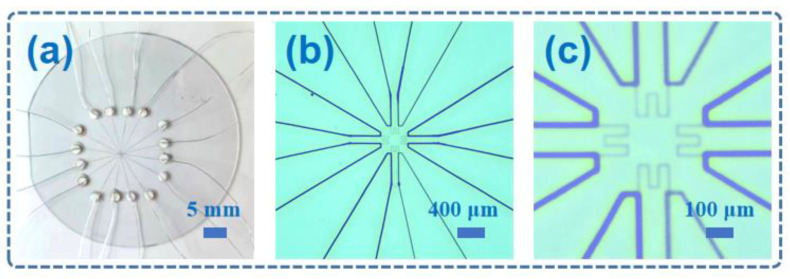
(**a**) The optical picture of the sensor without encapsulation at a scale of 5 mm. (**b**) The optical picture of the sensor without encapsulation at a scale of 400 μm. (**c**) The optical picture of the sensor without encapsulation at a scale of 100 μm.

**Figure 4 micromachines-14-01621-f004:**
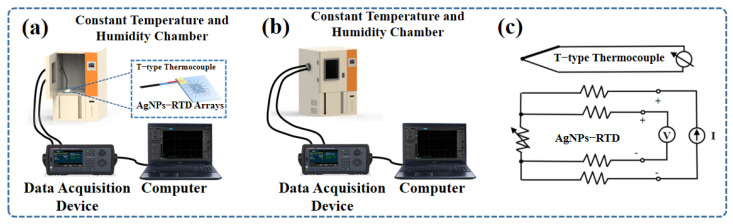
(**a**) Schematic diagram of the test platform. (**b**) Schematic diagram of the running test platform. (**c**) Four-wire configuration test circuit diagram.

**Figure 5 micromachines-14-01621-f005:**
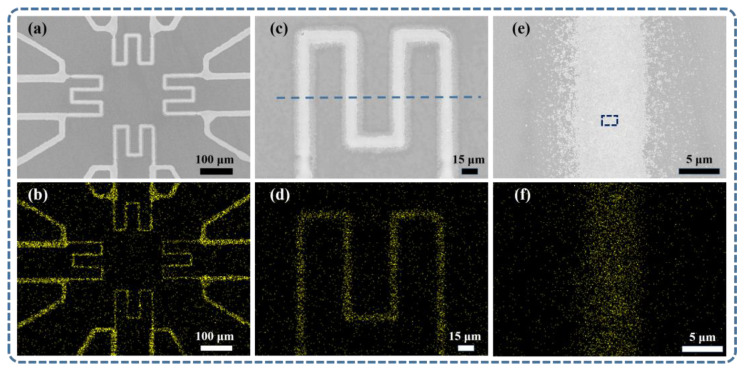
(**a**) SEM image of sensors array. (**b**) EDS image of Ag distribution of sensors array. (**c**) SEM image of the single sensor. (**d**) EDS image of Ag distribution of the single sensor. (**e**) SEM image of the line. (**f**) EDS image of the line.

**Figure 6 micromachines-14-01621-f006:**
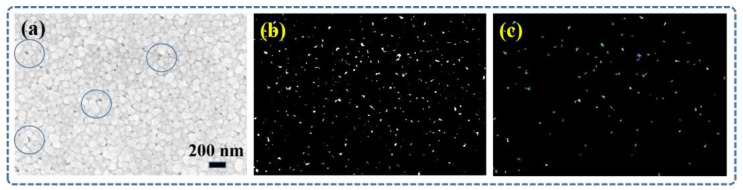
(**a**) SEM image of Ag nanoparticles distribution. (**b**) The binary image obtained after segmentation processing. (**c**) The image resulting from binary image calculation.

**Figure 7 micromachines-14-01621-f007:**
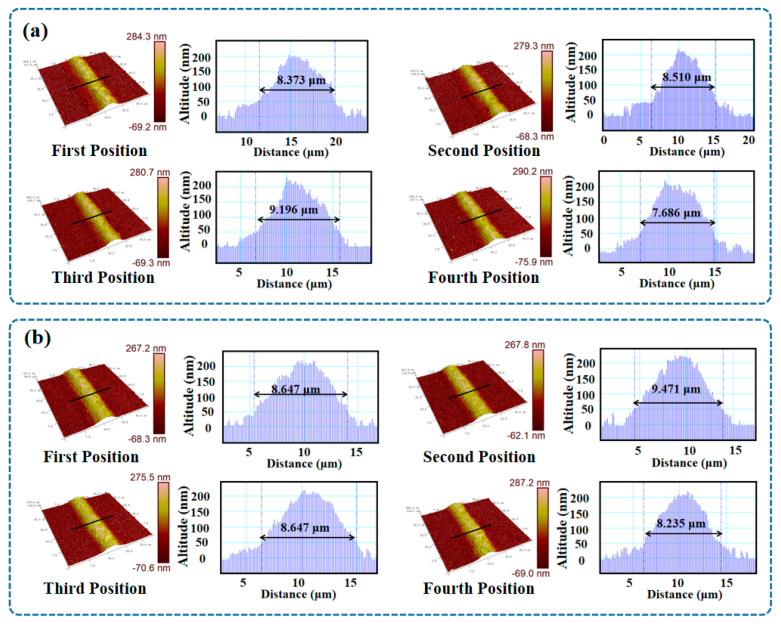
(**a**) AFM image and data of Sensor 1 in the sensors array. (**b**) AFM image and data of Sensor 2 in the sensors array. (**c**) AFM image and data of Sensor 3 in the sensors array. (**d**) EDS AFM image and data of Sensor 4 in the sensors array.

**Figure 8 micromachines-14-01621-f008:**
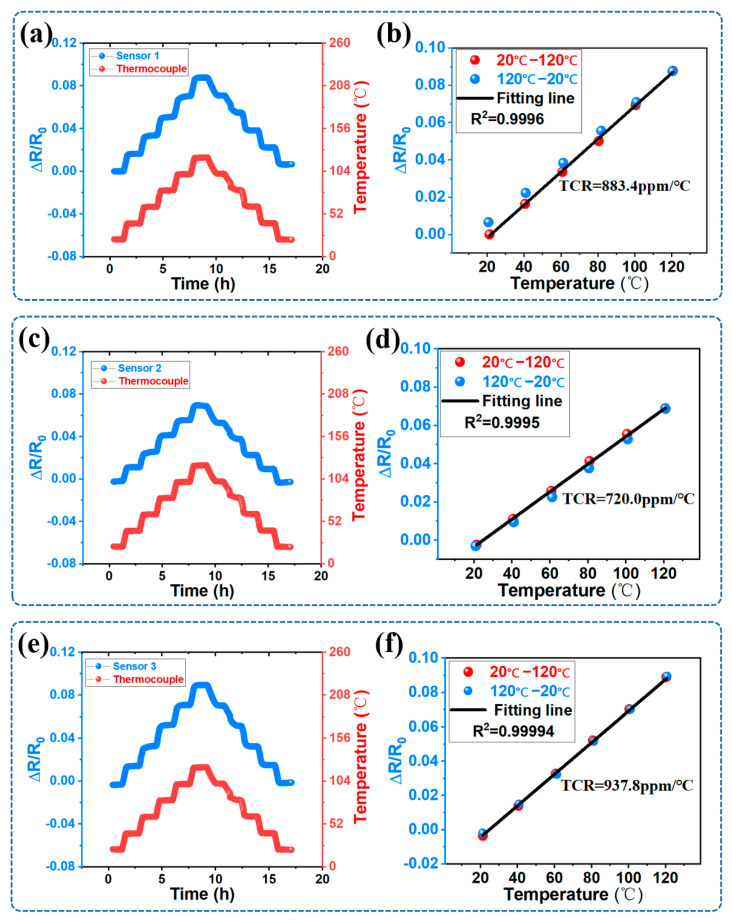
(**a**) The stepped temperature resistance test of sensor 1. (**b**) Resistance change rate with temperature variation of sensor 1. (**c**) The stepped temperature resistance test of sensor 2. (**d**) Resistance change rate with temperature variation of sensor 2. (**e**) The stepped temperature resistance test of sensor 3. (**f**) Resistance change rate with temperature variation of sensor 3. (**g**) The stepped temperature resistance test of sensor 4. (**h**) Resistance change rate with temperature variation of sensor 4.

## Data Availability

Not applicable.

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
