# Peer review of "Electrohydrodynamic Printed Ultra-Micro AgNPs Thin Film Temperature Sensors Array for High-Resolution Sensing"

_micromachines, 2023, doi:10.3390/mi14081621_

Round 1
Reviewer 1 Report
The work mainly deals with implementation of EHD technique in fabrication of arrayed temperature sensor. The work looks interesting. However, there are some points which are not obvious in the current form of the manuscript. Authors are encouraged to revise the manuscript following below comments.
1. Line 77-87, what is the significance of those lines in context to the present work?
2. It is not clear from the introduction part if authors are focusing on pattering fine lines OR EHD based temperatures sensors. The introduction part should be relevant to the reported work.
3. As the main context of the work looks like temperature sensors, authors must discuss the sensitivity of the previously reported temperature sensors along with fabrication for EHD based temperature sensors.
4. Moreover, some recently reported work on robust temperature sensors may be included, e.g.,
https://doi.org/10.1116/1.4884756
https://doi.org/10.1016/j.matpr.2022.10.077
https://doi.org/10.1016/j.vacuum.2020.109590
5. Line 220- How did authors verify the porosity and dense characteristics of the reported silver nanoparticles?
6. Line 260—How did the authors verify the oxidation? Please provide some evidence.
7. Moreover, what was the purpose of humidity and what was the value of humidity?
8. As the work is on temperature sensors, why authors didn’t provide the sensitivity of the sensors with goodness of fit?
9. Revise abstract and conclusion following above comments.
Check for grametical/typo mistake.
Reviewer 2 Report
The authors have successfully utilized the Electrohydrodynamic printing method to prepare thermistor-type AgNPs thin-film temperature sensors and performed preliminary calibration tests. The work is comprehensive and can be accepted with some modifications.
1. Please provide the full name of the term before using its abbreviated form, e.g. AgNPs, PDMS.
2. What is the minimum line width that can be achieved for AgNPs prepared based on Electrohydrodynamic Printed method? Is there a specific value to refer to?
3. Figures 6-9 as well as Figures 10-13 can be merged. Otherwise, the titles of the figures are exactly the same or similarly, which will let reader confused.
4. The four same sensors are supposed to be prepared using exactly the same process, but there is a big difference in the resistance change rate in Figs.10-13, what is the reason for this? The authors should be described and explained in the text. If it is due to the line width is less than 10 microns, in fact, according to the design of the structure, increase the width of the line is also able to achieve.
5. In the introductory section, there are a number of relevant papers for reference on the preparation of thin-film temperature sensors by screen-printing and MEMS technology. For example, Sensors and Actuators A: Physical, 2020, 315:112341; Microsystems & Nanoengineering, 2021, 7(1):42; International Journal of Extreme Manufacturing, 2023, 5(1):015601. Micromachines, 2021, 12(8): 924.
Reviewer 3 Report
The manuscript shows some results about AgNPs thin film sensor array. The topic is interesting. However, there are several points which must be addressed in the manuscript before further consideration. The manuscript can be reviewed after a major revision.
1. Figs. 6-9 should be re-integrated into one figure. On another hand, the description for them is too simple, 'As shown in Figure 6-9, the line width at the midpoint of the sensitive layer in the sensor array is less than 10 μ.' The images take up a lot of space in the manuscript, thus, more quantitative descriptions and conclusions can be added based on AFM results.
2. Similar to Comment 1#, Figs. 10-13 can be combined into one figure.
3. More discussion should be added into '3. Results and Discussion'. In the current version of the manuscript, it only include the preparation of the array, and SEM/electrical characterization. The reviewer encourages the authors to compare with similar results in other published papers for adding more discussion.
4. In Introduction, the authors focus in the field of the sensor. The synthesis and characterization of related materials for thin film arrays is also interesting, they can be added, e.g., 10.1016/j.vacuum.2022.111135, 10.1016/j.apsusc.2015.07.088.
5. Language should be carefully checked, e.g., on Page 6 Line 211, 'A.' in '3.1. A.Morphological and physical characterizations' should be removed; on Page 3 Line 108, 'holds' should be changed to 'hold'; on Page 3 Line 121, 'This' should be change to 'this'; on Page 3 Line 129, a period should be added before 'The performance'; on Page 6 Line 216, for 'four sensitive grids', its corresponding verbs should be pluralized; on Page 6 Line 230 & 232, 'Figure 6-7' and 'Figure 8-9' should be change to 'Figures 6-7' and 'Figures 8-9'. The reviewer did not list all language corrections required. This obviously reduces the readability of the manuscript.
See comment #5
Round 2
Reviewer 1 Report
Authors have provided sufficient responses to the raised queries. Based on those, the work may be accepted for publication.
Reviewer 3 Report
The manuscript has been improved, and I am glad to recommend its publication.